# Functional Materials Based on Cyclometalated Platinum(II) β-Diketonate Complexes: A Review of Structure–Property Relationships and Applications

**DOI:** 10.3390/ma14154236

**Published:** 2021-07-29

**Authors:** Ashanul Haque, Hani El Moll, Khalaf M. Alenezi, Muhammad S. Khan, Wai-Yeung Wong

**Affiliations:** 1Department of Chemistry, College of Science, University of Hail, Ha’il 81451, Saudi Arabia; h.elmoll@uoh.edu.sa (H.E.M.); k.alenezi@uoh.edu.sa (K.M.A.); 2Department of Chemistry, Sultan Qaboos University, P.O. Box 36, Al-Khod 123, Oman; 3Department of Applied Biology and Chemical Technology, The Hong Kong Polytechnic University, Hung Hom, Kowloon, Hong Kong, China

**Keywords:** cyclometalated, platinum(II), β-diketonate, opto-electronics, square planar

## Abstract

Square planar organoplatinum(II) complexes have garnered immense interest in the area of materials research. The combination of the Pt(II) fragment with mono-, bi- tri- and tetradentate organic ligands gives rise to a large variety of complexes with intriguing properties, especially cyclometalated Pt(II) complexes in which ligands are connected through covalent bonds demonstrate higher stability, excellent photoluminescence properties, and diverse applications. The properties and applications of the Pt(II)-based materials can be smartly fine-tuned via a judicious selection of the cyclometalating as well as ancillary ligands. In this review, attempts have been made to provide a brief review of the recent developments of neutral Pt(II) organometallic complexes bearing bidentate cyclometalating ligands and β-diketonate ancillary ligands, i.e., (C^N)Pt(O^O) and (C^C)Pt(O^O) derivatives. Both small (monomeric, dimeric) and large (polymeric) materials have been considered. We critically assessed the role of functionalities (ligands) on photophysical properties and their impact on applications.

## 1. Introduction

The last few decades have witnessed a great upsurge in the design and development of organometallic materials owing to their fascinating structures, intriguing properties and diverse applications [1,2]. Among a large number of organometallic materials, phosphorescent square planar Pt(II) complexes have gained particular interest [3,4,5]. This is indeed due to the inherent ability of heavy Pt(II) fragments to induce strong spin-orbit coupling (SOC), leading to intersystem crossing (ISC) as well as access to the spin-forbidden triplet radiative decay [6,7]. Consequently, Pt(II) complexes grafted in the main chains or side chains exhibit high triplet quantum yield (Φ_T_), long triplet excited state lifetime (τ_T_) and several other interesting photophysical properties [8]. In addition, due to the square planar coordination geometry around Pt(II) center, they often show molecular stacking supported by metallophilic and other noncovalent interactions (NCI) leading to altered emission features (such as excimeric emission).

In Pt(II) complexes, the photophysical and photochemical properties are highly sensitive to the type of the coordinating ligands. Through judicious selection of the ligands and their coordinating modes, the optical properties can be easily manipulated. Compared to the coordinate bonds, linkage of a metal to the organic core via one or more covalent bonds is highly beneficial as it imparts greater stability to the resulting complexes [9]. Studies suggest that the ligation of a strong electron-withdrawing cyclometalating ligand to a transition metal raises the energy of the metal-centered (MC) states to avoid nonradiative decay [3,10,11]. This leads to a significant impact on the emission parameters such as quantum yield (Φ) and lifetime (τ) values as well as color [12].

Cyclometalating ligands also offer an excellent platform for modulating NCI, which play a substantial role in determining the nature of photoluminescence (PL) [13]. Therefore, such ligands serve as excellent candidates for the design of new functional materials. Among different bi-, tri- and tetradentate ligands, the use of a bidentate ligand is particularly beneficial as it imparts high tunability of structural features and electronic properties to the Pt(II) complexes [14]. It has been demonstrated that the cyclometalated Pt(II) complexes induce greater SOC compared to other σ-bonded Pt(II) complexes such as platinaynes [15] and that the extent of SOC can be further enhanced by increasing the number of metal units [16].

In addition to the cyclometalating ligands and metallic core, the nature of ancillary ligand also plays a crucial role in the determination of physicochemical and photophysical properties [17]. Among the large variety of ancillary ligands, one very common and popular family is provided by β-diketonate derivatives [17,18,19,20,21]. The incorporation of structurally diverse β-diketonate ancillary ligands not only improves the physicochemical properties [18,19], but also alters molecular separation/stacking [17] induces liquid crystallinity [20] and governs the exciton generation and transport. It has been shown that the Φ and τ values can be readily improved simply by extending the ancillary ligand π-system [3,21].

Fascinated by the features of Pt(II) ion, bidentate cyclometalating organic core, and β-diketonate, a diverse range of small (monomeric, dimeric) and large (polymeric) complexes/materials have been developed in recent times [14]. These materials have been realized via smart selection of the C^N- or C^C*-cyclometalating ligands and β-diketonate ancillary ligands. Pt(II) complexes with β-diketonate ligands have been studied intensively, but they are not reviewed very often [14,22]. This motivated us to provide a brief and up-to-date review on neutral Pt(II) organometallic complexes bearing bidentate cyclometalating ligands and β-diketonate ancillary ligands, i.e., (C^N)Pt(O^O) and (C^C)Pt(O^O) derivatives (Figure 1). Following this introductory section, photophysical features of the Pt(II) complexes have been discussed. This is followed by application section in which we highlighted the organic photovoltaics (OPVs), organic light-emitting diodes (OLEDs), sensing, photocatalytic and other applications. Finally, concluding remarks are presented.

## 2. Structure–Property Relationships

As stated before, the optical properties and applications of cyclometalated Pt(II) β-diketonates are sensitive to the electronic nature, steric effect, and rigidity of cyclometalating ligands and β-diketonates. Most, if not all, of their properties can be tuned through manipulation of the main and ancillary ligands [23,24,25]. Besides, square planar configuration of Pt(II) complexes assists in avoiding detrimental aggregation caused quenching (ACQ), which is one of the most prominent causes of emission loss in transition metal complexes [26]. In the subsections below, we discuss the effects of different cyclometalating and ancillary ligands on the optical properties of Pt(II) complexes.

### 2.1. (C^C)Pt(O^O) Type Complexes

The last one-and-a-half decades witnessed a tremendous interest in the design and development of functional (C^C)Pt(O^O)-type complexes [14]. Most of the square planar complexes were based on N-heterocyclic carbene (NHC)-type cyclometalating ligands with β-diketonate derivatives. One of the main reasons behind selecting NHCs is their strong σ-donating tendency, high stability, and ability to tune the PL properties through structural modification. [27] Among NHCs, imidazole, pyrazole, and triazole-derivatives were the leading cores. Studies involving a variety of main and ancillary ligands show that there is no direct link between the optical properties and the substituents. The photophysical properties remain unchanged in some cases while they are enhanced in others. For example, complex **1** (Figure 2), based on imidazoe[1,5-*a*]pyridine-based Pt(II) complexes (**1**, Figure 2) with electron-withdrawing or donating moieties, did not show any significant effect of functional groups present over the cyclometalating/ancillary ligand on its PL properties [28]. However, a slight red-shifted absorption peak (~15 nm) was reported for **1** (R_1_ = OMe, R_2_ = H, R_3_ = *^t^*Bu, R_4_ = C_6_H_5_), which can be attributed to the extended conjugation.

On the other hand, researchers also noted that irrespective of the main ligand, ancillary ligand has a marked effect on the emission profile. For instance, when methyl (R = Me, λ_em_ = 457 nm, Φ = 0.41, τ_0_ = 9.2 μs in 2 wt% poly(methyl methacrylate) (PMMA)) group in complex **2** (Figure 2) is replaced by phenyl ring, the emission wavelength shifts to the red (λ_em_ = 520 nm, Φ = 0.78, τ_0_ = 4.8 μs in 2 wt% PMMA) [29]. However, the presence of other aromatic systems (R = mesityl (mes)/duryl (dur)) did not cause any significant shift. This unexpected behavior was attributed to the formation of more conjugated excited states in the case of the phenyl moiety, which can freely rotate [14]. There are also some examples where the emission is lost upon a slight change in the ancillary ligand. For example, replacing methyl (R = Me, λ_em_ = 463, 497 nm, Φ = 0.9, τ_0_ = 23 μs in 2 wt% PMMA) by trifluoromethyl (CF_3_) group in complex **3** (Figure 2) was found to be detrimental for emission [30,31]. As noted in the previous example, and in this case as well, complexes containing phenyl and mes showed a red-shifted emission (λ_em_ = 465 nm for R = *^t^*Bu → λ_em_ = 530 nm for R = C_6_H_5_, Table 1) and improved Φ (Φ = 0.83 for R = *^t^*Bu → Φ = 0.91 for R = mes, Table 1), respectively.

Pinter et al. [32] studied the effect of introducing nitrogen into the backbone of the NHC and prepared a new class of compounds containing different auxiliary ligands (**4**, Figure 2). Interestingly, complexes containing the new heterocyclic ligand precursors presented an important influence on the emissive properties (Table 1) and aggregation behavior. In addition, the use of bulky ancillary ligand avoided molecular aggregation leading to excimeric emission. The combination of NHC containing pyridine heterocycle and bulky ancillary ligands allowed for remarkable emissive properties (Φ = 0. 91 and 0.88, for R = mes and dur, respectively) to be reached.

The photophysical analysis of 3-methyl-1-phenylimidazole-based complexes **5** (Figure 2) revealed that the Φ and τ_0_ values could be improved by introducing bulkier auxiliaries [17,31]. For example, it was noted that the introduction of mes or dur substituted β-diketonate [17] in **5** (R = Me) [31] led to a red shift of the emission band, >10–12× increment in the Φ and decay times (τ_0_ ~ 3 μs) (Table 1). To compare the effect of the saturated backbone of the NHC ligand on the photophysical properties, Stipurin and Strassner [11] reported Pt(II) complexes **6** (Figure 2) based on 1-aryl-3-methyl-imidazolinylidene ligands. These thermally stable complexes were found to absorb in the 280–450 nm region and emit between 470–490 nm with Φ = 0.24–0.7 and τ_0_ = 5–25 μs. The wavelength maxima showed a slight dependency on the cyclometalating ligand. [11] Complex **7** (Figure 2) exhibits wavelength-dependent features with emission in the blue to green region of the visible spectrum when excited at 330 nm [33]. In such complexes, the presence of electron-withdrawing substituents in the 4-position of the cyclometalated ring confers very high Φ (Table 1). On the other hand, chloro-substituents at 3- and 5-positions of the cyclometalated ring have detrimental effect on the emission efficiency. The substitution effect was also observed on the formation of tellurium-based clusters. Electron-withdrawing substituents at 4-position favored the formation of the self-assembled 2D structure, whereas chloro-substituents at the 3- and 5-positions leads to a discrete cluster. The inclusion of Tl(I) between the Pt(II) shows no beneficial effect on the Φ values [33].

There also exists a way of tuning the physical properties by changing the binding mode of NHC ligands. Inspired by their previous studies that highlighted the donor ability of using 1,2,3-triazol-5-ylidene, Strassner and coworkers [34] reported the first example of cyclometalated Pt(II) complexes with abnormal coordination mode of the main imidazolinylidene ligands (**8**, Figure 2). The reported complex showed ligand (cyclometalated and ancillary)-dependent absorption (λ_abs_ = 225–450 nm in DCM solution) and emissions (λ_em_ = 523–585 nm in 2 wt% in PMMA, Figure 3) profile with strong phosphorescence at room temperature and long-lived excited state. The emission was attributed to a strongly metal-perturbed intra-ligand/metal-to-ligand charge-transfer (ILCT/MLCT) process.

1,3,4-Triphenyl-1,2,4-triazole is a well-known ligand with mono-[35] and bidentate [36] coordination modes. It was found that the cyclometalated complex **9** (Figure 2) containing this ligand and acetylacetonate (acac) as an ancillary showed a phosphorescent emission with Φ = 0.42 and τ_0_ = 35.6 μs (Table 1). On the other hand, structurally close analogue **10** (Figure 2) [37] with mes/dur unit showed higher values of Φ (2×) attributed to the steric protection in the excited state by the bulky β-diketonate ligands. Contrary to this study, when the *C*-donor (phenyl) was replaced by a dibenzo[b,d]furan-4-yl (DBF) moiety in Pt(II) complexes coordinated to 1,2,4-triazol-5-ylidene derivatives (**11**–**12**, Figure 2) [23], the PL properties showed dependency on both the DBF as well as the type of ancillary ligands. These complexes, which show multiple absorption peaks in the region of 250–390 nm, were blue-green emissions in the PMMA matrix (Φ = 0.41–0.87 and τ_0_ = 6–26 μs). Due to the differences in the arrangement of the DBF unit, complex **12** (R = Me, mes) had blue-shifted emission compared to **11** (R = Me, mes). One of the complexes (**11**, R = Me) also exhibited emission in neat film due to the formation of multimolecular complex while **12** (R = mes) showed emission due to monomeric species. This observation was attributed to the presence of a sterically demanding dimesitylmethanato ligand, which efficiently suppresses molecular stacking. On comparing **3** and **11** (R = Me), which have different NHCs, we find somewhat different PL parameters, the former having higher Φ value but almost similar lifetimes (Table 1).

Very recently, using one of the DBF isomers, the same group prepared 1,2,3-triazolylidene ligand-based Pt(II) complexes (**13**, Figure 2) [38]. These compounds showed absorption in the UV region with onset of absorption at ~425 nm and emission maxima ~490 nm (Table 1). As expected, these complexes exhibit high Φ (0.67–0.78) and τ_0_ = 17.7–20.5 µs in the solid matrix than in the solution (Table 1). It was concluded that the ancillary ligand had no major effect on the emission and redox properties.

**Table 1 materials-14-04236-t001:** Photoluminescence data of some selected complexes (wt% in PMMA) at room temperature.

Complex	λ_exc_	λ_em_ ^c^	Φ	τ_0_ (μs)	Ref.
**3** (R = *^t^*Bu) ^a^	355	465	0.83	27.4	[30]
**3** (R = C_6_H_5_) ^a^	370	530	0.51	4.7	[30]
**3** (R = mes) ^a^	335	466	0.91	18.9	[30]
**3** (R = Me) ^a^	355	463, 497	0.90	23	[31]
**4** (R = Me) ^a^	340	458, 547	0.64	4.8	[32]
**4** (R = *^t^*Bu) ^a^	340	456	0.74	5.8	[32]
**4** (R = mes) ^a^	340	472	0.91	2.9	[32]
**4** (R = dur) ^a^	355	466	0.88	3.3	[32]
**5** (R = Me) ^a^	320	441	0.07	-	[31]
**5** (R = mes) ^a^	320	482	0.82	3.1	[17]
**5** (R = dur) ^a^	320	479	0.72	3.3	[17]
**6** (R_1_ = H, R_2_ = Me) ^a^	305	470	0.29	23	[11]
**6** (R_1_ = H, R_2_ = mes) ^a^	305	486	0.71	5	[11]
**6** (R_1_ = Cl, R_2_ = Me) ^a^	305	473	0.38	25	[11]
**6** (R_1_ = Cl, R_2_ = mes) ^a^	305	476	0.7	7	[11]
**6** (R_1_ = Me, R_2_ = Me) ^a^	305	481	0.24	23	[11]
**6** (R_1_ = Me, R_2_ = mes)	305	490	0.42	8	[11]
**7** (R_1_ = CN, R_2_ = H) ^b^	330	470	0.98	-	[33]
**7** (R_1_ = COOEt, R_2_ = H) ^b^	330	474	0.93	-	[33]
**7** (R_1_ = H, R_2_ = Cl) ^b^	330	450	0.04	-	[33]
**9** ^a^	355	441, 465	0.42	35.6	[37]
**10** (R = mes) ^a^	330	470	0.82	7.2	[37]
**10** (R = dur) ^a^	330	467	0.87	8.5	[37]
**11** (R = Me) ^a^	370	474, 506, 542	0.70	26	[23]
**11** (R = C_6_H_5_) ^a^	360	526	0.41	6	[23]
**11** (R = mes) ^a^	355	473, 505, 541	0.76	26	[23]
**12** (R = Me) ^a^	330	446, 474	0.71	24	[23]
**12** (R = C_6_H_5_) ^a^	370	524	0.45	7	[23]
**12** (R = mes) ^a^	340	445, 474	0.87	10	[23]
**13** (R = Me) ^a^	380	490	0.67	20.5	[38]
**13** (R = mes) ^a^	380	488	0.69	17.9	[38]
**13** (R = Dur) ^a^	380	488	0.78	17.7	[38]
**14** (R_1_ = Bn, R_2_ = Me, R_3_ = H) ^a^	330	432	0.13	15	[39]
**14** (R_1_ = Me, R_2_ = mes, R_3_ = H) ^a^	340	478	0.82	4	[39]
**14** (R_1_ = Me, R_2_ = mes, R_3_ = OMe) ^a^	355	477	0.82	5	[39]
**14** (R_1_ = Me, R_2_ = mes, R_3_ = Me) ^a^	355	475	0.82	3	[39]
**14** (R_1_ = Bn, R_2_ = mes, R_3_ = H) ^a^	370	477	0.72	5	[39]
**14** (R_1_ = Bn, R_2_ = mes, R_3_ = OMe) ^a^	370	478	0.70	7	[39]
**14** (R_1_ = Bn, R_2_ = mes, R_3_ = Me) ^a^	340	471	0.81	5	[39]
**15** (R_1_ = Bn, R_2_ = Me) ^a^	330	455	0.35	15	[39]
**15** (R_1_ = Me, R_2_ = mes) ^a^	345	477	0.82	4	[39]
**15** (R_1_ = Bn, R_2_ = mes) ^a^	355	477	0.83	4	[39]
**16** (R = Me) ^a^	380	538	0.47	25.4	[40]
**16** (R = mes) ^a^	380	537	0.53	22	[40]
**16** (R = dur) ^a^	380	537	0.54	22.6	[40]

^a^ = 2 wt%, ^b^ = 5 wt%, ^c^ = Wavelength at the emission maximum.

Another study revealed that the emission behavior of Pt(II) complexes is independent of the position of the *N* atom in the backbone of the NHC unit but depends on the auxiliary ligand (O^O). A comparative study of Pt(II) complexes possessing 1-phenyl-1,2,4-triazol-5-ylidene or 4-phenyl-1,2,4-triazol-5-ylidene as C^C ligand and acac or 1,3-bis(2,4,6-trimethylphenyl)propan-1,3-dionato (mesacac) as ancillary ligands (**14**–**15**, Figure 2) was carried out by Strassner and coworkers [39]. It was reported that the complexes show very weak emissions in solution but are strongly emissive (λ_em_ = 432–478 nm) with high quantum efficiencies (Φ = 0.13–0.82) and short decay times (τ_0_ = 3–15 μs) with mesityl substituted derivative (mesacac) ancillary ligands in PMMA matrix (Table 1). To underpin whether the extension of a conjugated system or annulation to the NHC ligand has a pronounced effect on the optical properties, cyclometalated Pt(II) complexes with annulated 1,2,3-triazolylidene-based ligand (**16**, Figure 2) have been reported [40]. The emission properties of these complexes were nearly similar at room temperature with emissions in the yellow region (537–539 nm) with Φ = 0.47–0.54 (Table 1). The use of sterically demanding β-diketonate ligands led to slightly increased efficiencies compared to the parent acac complex.

Among the N-containing NHC donors, thiazole as C-cyclometalating unit has also been tested. To determine the impact of electronic effect, 4- and/or 5-position of the 1,3-thiazole moiety as well as *N*-aryl moiety were substituted [41,42]. Complexes **17** (R_1_ = Me, C_6_H_5_, C_4_H_4_; R_2_ = H, COOMe, Me, C_6_H_5_, C_4_H_4_; R_3_ = Me; R_4_ = R_5_ = R_6_ = H, Figure 4) with functionalized thiazole ring (different substituents in the 4- and 5-position of the thiazole heterocycle) showed broad absorption peaks at around ~275, 300 and 360 nm and bluish-green phosphorescence (497–522 nm) at room temperature in 2 wt% PMMA. The emission in such complexes was suggested to be ^3^MLCT/^3^ILCT admix nature. It is interesting to note that the absorption intensity, wavelength, quantum yields, and decay lifetimes did show variation with the inductive and mesomeric effects of the substituents over thiazole moiety; however, the maximum emission wavelength was found to be almost independent. Overall, the presence of methyl and ester groups was beneficial in terms of Φ and τ_0_.

On the other hand, when substituents on the *N*-phenyl ring were modified (**17**, R_1_ = R_2_ = R_3_ = R_4_ = Me; R_5_ = H, Br, OMe, CN, Me, COOMe; R_6_ = H, Me, Figure 4) [42], a strong absorption between 250–280 nm was observed with some additional peaks ~360–370 nm. The PL study revealed that all complexes emit in a very narrow range (508–526 nm; blue-green) with high Φ (0.33–0.79) and τ_0_ (8.1−21.4 μs) values. The emission wavelength maximum was almost independent of electron-donating or electron-withdrawing groups at the cyclometalating aryl ring. However, different shapes of the emission spectra were indicative of different underlying emission processes. Like the previous example, the presence of the ester group over the phenyl ring was beneficial as complex **17** (R_1_ = R_2_ = R_3_ = Me; R_4_ = R_6_ = H; R_5_ = COOMe, Figure 4) showed high Φ (0.79 in 2 wt% PMMA) and low τ_0_ (8.11 µs in 2 wt% PMMA), which retained high Φ (0.68 in the 100%) in film.

Another study reflected the effect of variation of auxiliary ligand on PL properties. Except for complex **17**, which beared an electron-withdrawing substituent (R_1_ = R_2_ = Me; R_3_ = CF_3_; R_4_ = R_5_ = R_6_ = H, Φ = 0.01), all other complexes (R_1_ = R_2_ = Me; R_3_ = Me, *^t^*Bu, C_6_H_5_, mes, dur; R_4_ = Me, C_6_H_5_; R_5_ = R_6_ = H, Figure 4) emitted in the green to yellow region (508–526 nm) with high Φ (0.17–0.66) and low τ_0_ (5.4–13.7 µs) in 2 wt% PMMA) [18]. In these complexes too, the emission maxima were independent of the β–diketonate ligand. Anderson and coworkers [43] reported a comparative analysis of five- and six-membered Pt(II) complexes **18**–**20** (Figure 4) based on thiophene (Th) / benzothiophene (Bth) cores. The reported complex exhibited absorption in the 325–450 nm region and emission bands within the 430–530 nm region from a ^3^MLCT excited state. Whereas imidazole-containing complexes display a broad monomeric emission, benzimidazole showed two bands due to excimeric emission. In addition, complexes **18** and **19** had lifetimes between 241–959 μs, while **20** had 350–989 μs. When compared between 5/6-membered metallacycles, it was noted that the Bth-containing complexes displayed a lower magnitude red-shifted emission band compared to their Th-containing analogues. The opposite effect is noted in the five-membered metallacycles, where the benzothiophene-containing analogues displayed a greater, red-shifted emission band compared to their thiophene-containing analogues.

### 2.2. (C^N)Pt(O^O) Type Complexes

Compared to C^C* cyclometalated Pt(II) complexes, the C^N analogues exhibit a wider range of properties/applications. Various complexes of the type (C^N)Pt(O^O) featuring 2-phenylpyridine (2-ppy), 2-thienylpyridine, 2-phenylbenimidazole, etc. cyclometalating ligands supported by diketonate derivatives have been assessed recently. A minor change in the functional group or position has a remarkable effect on their PL properties [44]. It has been demonstrated that the topology/connectivity of metal to ligand influences the photophysical properties, and this has been exploited to realize long-lived ^3^IL or dual (fluorescence and phosphorescence) emission [45,46] and charge transfer (CT) [47]. Fluorinated substituents are well known for altering the electron distribution and providing steric protection around the metal center. The installation of a strong acceptor -CF_3_ on the cyclometalated 2-ppy at the 5-position of the pyridine ring resulted in a red shift of the emission compared to complex based on nonfunctionalized 2-ppy [48]. In the literature, most of the bidentate C^N main ligands are based on 2-ppy derivatives as they can be easily functionalized to target desired properties. Zhong and coworkers [49,50] demonstrated that a minor variation in the ppy core affects the molecular packing and emission properties. They introduced one or two fluorine substituents in the ppy core to realize complex **21** (R_1_ = R_2_ = C_6_H_5_; R_3_ = Me; R_4_ = R_5_ = R_6_ = H, F, Figure 5) having a similar emission profile in solution but was different (green to orange) in the solid state.

All complexes showed an admix of LLCT and MLCT transitions in the region of 330 nm to 450 nm [49]. However, the maximum emission wavelengths were the same for all complexes in solution, Φ and τ varied according to the number of F-atoms. In general, complex **21** (R_1_ = R_2_ = C_6_H_5_; R_3_ = Me; R_4_ = R_6_ = F; R_5_ = H, Figure 5) having two F atoms showed higher τ (1.02 µs) and Φ (0.16) values in solution. Moreover, microcrystals demonstrated different luminescence properties than macro/large crystals (Figure 6). Microcrystalline emission (up to Φ = 0.51) was also reported in 2-phenylbenzimidazole-based Pt(II) complexes bearing acac ancillaries [26].

Several researchers kept the acac as ancillary ligand and functionalized the main ligand to study both the electronic and steric effects. Bearing this in mind and to underpin how the position of a fluorophenyl substituent affects the structural, photophysical properties and applications (oxygen sensing), Xing et al. [51] reported complex **21** (R_1_ = R_2_ = Me; R_3_ = R_4_ = H; R_5_, R_6_ = 4-C_6_H_4_F, 3,4-C_6_H_3_F_2_, 3,4,5-C_6_H_2_F_3_, Figure 5) containing 2-ppy with a fluorophenyl substituent at the *para* or *meta* position of the phenyl ring. As observed in similar complexes, the reported fluorophenyl containing complexes showed no metallophilic (Pt··Pt) interaction in the solid state. Moreover, compared to the parent complex Pt(ppy)(acac), both *meta* and *para* complexes showed red-shifted absorption and emission maxima (λ_Pt(ppy)(acac)_ < λ*_meta_*
_complexes_ < λ*_para_*
_complexes_). This can be attributed to the presence of strong electron-withdrawing fluorine substituents, which lowers the energy level of frontier molecular orbitals. Moreover, a noticeable difference was also found in the τ_0_ (1.39–4.11 μs for *para* and 1.03–2.23 μs for *meta*) and Φ values (0.15–0.22 for *para* and 0.14–0.22 for *meta*). It was noted that compared to Pt(ppy)(acac) (τ_0_ = 2.6 μs), the lifetime value reduced when the number of fluorine atom increases (except for *para*). The role of fluorine, phenyl, and 2,4-difluorophenyl substituents **21** (R_1_ = R_2_ = Me; R_3_ = R_5_ = H; R_4_ = H, F; R_6_ = F, C_6_H_5_, 2,4-C_6_H_3_F_2_, Figure 5) on the photophysical and sensing properties were also studied by Liu and coworkers [52] These complexes also showed very similar photophysical data to earlier discussed complexes. For instance, emission maxima between 467–526 nm, τ_0_ = 1.24–4.56 μs and Φ = 0.04–0.24 was noted for the complexes. Complexes bearing phenyl or fluorophenyl substituents showed red shifted emission and higher quantum yield compared to the Pt(ppy)(acac) and other fluorinated complexes. Considering the properties and features of such complexes, it can be concluded that changing the substituent/topology (connectivity) affects the electronic distribution leading to the change in photophysical properties. Okamura et al. [53] reported the properties and OLED application of Pt(II) complexes bearing 5′-benzoylated 2-ppy ligands with or without fluorine substituent(s) **22** (Figure 5). The combination of benzoyl and fluorine substituents had a noticeable impact on the τ_0_ (0.3–1.37 μs), Φ (0.06–0.28) and emission spectra (465–479 nm and 497–513 nm in solution). In addition, the presence of fluorine and carbonyl groups led to enhanced excimeric emission. As noted earlier, Φ value varies inversely with the number of fluorine atoms. It was suggested that the electronic effect of the carbonyl was the main role-player rather than π-extension factor.

The modulation of luminescence properties exploiting NCI in organic systems is an intriguing strategy. However, such observation is limited to organometallic systems. An account of modulated PL properties and application via intermolecular interactions can be found in a recent review by Klein and coworkers [13]. Kang et al. [54] noted that the introduction of a bulky trimethylsilyl (TMS) group at the 4- or 5 positions of the *N*-donating pyridine ring **23** (Figure 5) is an effective strategy to endow intermolecular interaction and modulate structural as well as photophysical properties. For example, it was noted that the complex **23** (R = none) forms stacked dimer via Pt(II)···π interactions while complexes **23** (R = 4/5-TMS) prefer dimeric structure supported by Pt···Pt and π−π interactions (Figure 7). Complexes containing TMS exhibit red-shifted absorption and emission profiles than the one without TMS. A comparison of the frontier molecular orbital energy level indicated that the introduction of a TMS group lowered the HOMO level (with little impact on LUMO level) and increased the energy gap (Eg) compared to Pt(ppy)(acac) and other related fluorinated systems. The presence of strong intermolecular interaction also improved the quantum yield (Φ = 0.44 for **23** (R = none) and 0.60–0.65 for **23** (R = 4/5-TMS) in DCM solution) of the complexes and assisted in excimeric emission. Kukushkin and coworkers [55] adopted another strategy to enhance the luminescence properties. They found that the 1:1 co-crystal prepared using π hole-donating electron-deficient arene systems (viz. perfluorinated arenes) and Pt(II) complex such as Pt(ppy)(acac) or Pt(pbz)(acac) (where pbz= 2-phenylbenzothiazole) exhibit π-hole···dz^2^(Pt^II^) interactions. Owing to such interactions, enhanced phosphorescence was reported in the solid states. For example, Pt(ppy)(acac) and octafluoronaphthalene exhibit much better PL properties (Φ = 0.11, τ_0_ = 8.42 μs) than the Pt(II) complex alone (Φ = 0.03, τ_0_ = 0.58 μs) in the solid state. Overall, the work opens up a new dimension for tuning the PL properties via altering supramolecular properties.

Among various molecular engineering strategies [6,7], one established way is to create an intramolecular donor–acceptor (D-A)-type interaction. Such interaction is often used to achieve low Eg materials with near infra-red absorption/emission features. Wong and coworkers [56] found that the D-A interaction can be achieved via adjusting coordination position with isomeric ligands. Isomeric complexes **24** and **25** (X, Y = C/N, Figure 5) displayed excellent thermal stability, photophysical ($E_{g,\;film}^{op.}=1.72-2.23\;\mathrm{eV}$ with $\lambda_{\mathrm{onset}}^{\mathrm{film}}=556-720\;\mathrm{nm}$, $\lambda_{\mathrm{em}}^{\mathrm{sol}}=590-761\;\mathrm{nm},$ τ_0_ = 0.11–1.09 μs and Φ = 0.6–2.2), electrochemical ($E_{ox.}^{1/2}=0.86-1.01\;V\;\mathrm{vs}\;\mathrm{Ag}/\mathrm{AgCl})$, and electroluminescent (EL peaks between 626–826 nm and EQE = 0.04–0.49%) properties.

Usuki and coworkers [57] reported a comparative property of complexes **26** (Figure 5). Due to the presence of bulky side chains (which reduced π−π interactions) complexes **26** (R = -C(Me_2_)(C_6_H_5_), -Si(Me_2_)(C_6_H_5_), -Si(Me)(2,5-Me_2_C_6_H_3_)(2-OMe-C_6_H_4_) exhibit unique emission features in the solid and solution states. For instance, 2**6** (R = H) showed weak emission in the solid state (due to π−π stacking in the solid state), while other complexes exhibited bright luminescence in the solid state with good emission efficiency (Φ_P_ = 0.33–0.49, τ_0_ = 1.80–14.6 μs). Overall, this work demonstrated that the introduction of a side chain containing quaternary carbon or silicon atoms into the Pt(thpy)(acac) skeleton promoted solid-state emission without impairing either the emission efficiency or color. To study the effect of the inclusion of chalcogenides (O, S, Se, Te) in Pt(II) complexes, Wu and coworkers [58] reported complexes **27** (Figure 5). These complexes exhibit an intense π → π* band below 400 nm and a low energy weak band of singlet and triplet CT nature in the visible. It was noted that instead of a variation of absorption and emission wavelengths in the order O → S → Se → Te (due to enhanced electron-donating ability), S-containing complex showed some deviation. For instance, **27** (X = S) (λ_em_ = 611 nm and 664 nm) showed blue-shifted emission peaks as **27** (X = O) (λ_em_ = 616 nm & 674 nm), **27** (X = Se) (λ_em_ = 621 and 674 nm) and **27** (X = Te) (λ_em_ = 634 nm and 690 nm). High aromaticity of the Th ring was suggested as the major factor behind this observation.

In addition to the above-mentioned mononuclear complexes, some excellent di- and trinuclear cyclometalated Pt(II) complexes have also been reported in recent times. This includes bimetallic complexes (homo-/hetero-multi-nuclear metal complexes) with unique bridging ligands [59]. Figure 8 illustrates some examples of homo- and heterobimetallic complexes reported recently, while Table 2 depicts the absorption and emission profile of some selected mono- and bimetallic Pt(II) cyclometalated complexes. The inclusion of the second metal atom not only enhances the SOC and adjusts the emissive triplet states, but also improves their properties and performance [16]. For example, complex **28** (Figure 8) showed a dual phosphorescence emissive character at room-temperature with increased Φ compared to that of its monometallic counterpart **29** (Figure 8) [60]. Although there was no major effect on the light-harvesting ability upon the inclusion of second metal, the effect on the emission properties was pronounced. The monometallic complex showed fluorescence/phosphorescence emission at 498/698 nm with a minor peak at 600 nm (τ_0_ = 6 μs, Φ = 0.09).

Complex **28** showed slightly longer τ_0_ [8.36 μs (700 nm) and 12.81 μs (600 nm)] than those of the mono-nuclear complex **29** [6.53 μs (700 nm), 6.06 μs (600 nm)]. In addition, a higher Φ value was reported for the bimetallic complex **28** (0.17 at 697 nm and 0.07 at 600 nm] than mono-metallic **29** [0.09 at 698 nm] counterparts. In the case of **28**, a similar emission profile was observed with the enhanced emission band at around 600 nm, but reduced fluorescence emission, an indication of dual phosphorescence emission characteristics. The enhancement of the band at around 600 nm was attributed to the second Pt(II) center, which facilitated the formation of the ^3^LMCT state that lies close to either the original ^3^MLCT state or the ^3^ILCT state. Other pyrene-based high-efficiency NIR-emitting mono- and bimetallated Pt(II) complexes were reported by Liu and coworkers [61]. Using a shoulder-to-shoulder type A-D-A ligand, complex **30** (Figure 8) was synthesized and compared with the monomeric counterparts **31** (Figure 8). A similar enhanced intramolecular SOC effect along with improved photophysical properties was observed. Unlike the previous example, the introduction of a second metal ion improved the light-harvesting ability and caused red shift and lowering of the band gap (E_g_ = 2.15 and 1.88 eV). Similarly, emission studies reveal bathochromic shift of the emission maxima to deep red region by 74 nm, with Φ = 16.94%, τ = 0.39 μs.

A deep red-NIR emission was also noted in complex **32** (Figure 8). Compared to the mono-metallic complex **33** (Figure 8) (λ = 618 nm, Φ = 2.42%, τ = 0.37 μs), the bimetallic complex **32** showed 112-nm red-shifted NIR emission (λ = 730 nm, Φ = 0.77%, τ = 0.26 μs, Table 2) [62]. A red phosphorescence (λ = 615 nm, Φ = 0.85, τ = 0.64 µs) was also reported for complex **34** (Figure 8) [63]. In this complex, the Ir(III) centre plays the dominant role than Pt(II) in modulating triplet level. Interestingly, this complex reveals very high radiative rate (*k*_r_ =1.33 × 10^6^ s^−1^), better than various Pt(II) and Ir(III) complexes. Wong and coworkers [64] demonstrated that the emission can be shifted to red and a photoluminescence quantum yield can be increased by increasing the number of metallic fragments **35**–**37** (Figure 8). Moreover, by incorporating more Pt(II) centers, the phosphorescence emission can be greatly enhanced, with the photoluminescence quantum yield (PLQY) increasing from 0.24 to 0.74 in solutions.

**Table 2 materials-14-04236-t002:** Absorption and emission profiles of some selected mono- and bimetallic Pt(II) cyclometalated complexes.

S. No.	PL Parametres				Ref.
	λ_abs_ ^a^ nm, (ε × 10^−4^ M^−1^ cm^−1^)	λ_PL_ ^a^ (nm)	Φ ^a^ (%)	τ_0_ _(µs)_	
**30**	601 (0.44), 435 (1.64), 383 (7.75), 318 (3.79), 275 (4.27)	704	16.94	0.39 ^b^	[61]
**31**	504 (0.48), 364 (9.09), 329 (4.73), 285 (5.43)	630	21.44	0.45 ^b^	[61]
**32**	570 (6.6), 535 (6.1), 450 (1.3),424 (1.2), 392 (2.8),375 (2.0), 310 (3.4), 286 (4.0)	730	0.77	0.26 ^c^	[62]
**33**	413 (0.97), 369 (2.15), 352 (1.81), 325 (2.22), 270 (3.06), 280 (2.87)	618	2.42	0.37 ^c^	[62]

^a^ in CH_2_Cl_2_ solution, ^b^ in degassed toluene, ^c^ in degassed CH_2_Cl_2_.

## 3. Applications

### 3.1. Organic Photovoltaics (OPVs)

Photovoltaic cells or photoelectric cells are electrical devices that convert light energy to electrical energy through a photovoltaic effect [65]. In the quest for renewable energy sources, researchers have been investigating the organic, inorganic and organic–inorganic hybrid compounds [66]. Owing to their cost-effectiveness, light weight, high flexibility and ease of processing, π-conjugated materials have attracted the attention of researchers to explore the possibility of application in OPV devices [67,68,69]. Although still lagging behind the value (efficiency) required for commercialization, recent decades saw an excellent advancement in the field of organic photovoltaics utilizing small molecules to large polymers [7,70,71]. Organic π-conjugated materials coordinated with heavy metals were also found to display enhanced generation of triplet excited states yielding moderate to high efficiency [72,73,74].

In general, to efficiently convert light to electricity, a material should possess balanced light absorption, mobility of charges and the diffusion length of excitons [75]. Researchers are working towards one or more of these aspects to develop materials with high efficiency. For example, it has been demonstrated that the combination of a π-conjugated D-A units result in low E_g_ materials with absorption spanning the visible region of the solar spectrum. Similarly, tuning the planarity, introducing NCI, enhancement of intermolecular π−π interactions and self-assembly etc. are some other excellent approaches to obtain high-performance materials [7].

Recently, a new structural design strategy of random terpolymers has been found to be an effective approach for designing new conjugated polymer materials. Unlike the conventional D−A alternating copolymers, terpolymers are comprised of three different monomeric units as the repeating group in the conjugated polymer backbone. Attempts have also been made to realize OPV based on the small-to-large donor sas well as acceptor active layers using cyclometalated Pt(II) complexes. Polymers containing cyclometalated Pt(II) units in the main chain and side chain have been assessed. Fréchet and coworkers [76] demonstrated that thiophene- and fluorene-containing polymers **38** (Figure 9) are excellent alternatives to platina-ynes, possibly due to the greater extent of SOC in cyclometalated complexes [15]. They reported that the thiophene-containing cyclometalated polymer exhibits much better performance (*J*_sc_ = 5.3 mA/cm^2^, V_oc_ = 0.65 V, FF = 0.37, PCE = 1.29%) than fluorene containing polymer (*J*_sc_ = 3.5 mA/cm^2^, V_oc_ = 0.38 V, FF = 0.30, PCE = 0.4%). This difference in the performance was attributed to the better overlap with the solar spectrum of the former. It should be noted that this value (PCE = 1.29%) is much better than several poly(platina-ynes) [6,7].

Three years later, Cheng and coworkers [77] reported an indacenodithiophene-based π-conjugated polymer **39** (Figure 9). This new polymer showed much better performance (*J*_sc_ = 4.7–7.7 mA/cm^2^, V_oc_ = 0.78–0.79 V, FF = 0.36–0.49, PCE = 1.7–2.9%) than the earlier reported polymer. The performance, which was the highest at that time, was realized using phenyl-C_71_-butyric acid methyl ester (PC_71_BM) acceptor (in contrast to the phenyl-C_61_-butyric acid methyl ester (PCBM) used in an earlier study). The incorporation of Pt(II) fragment red-shifted the absorption band, lowered the frontier orbital energy levels and improved the solar cell performance compared to metal-free polymers (*J*_sc_ = 3.5–4.0 mA/cm^2^, V_oc_ = 0.69–0.80 V, FF = 0.32–0.44, PCE = 0.9–1.2%). A further change in the cyclometalated Pt(II) “auxochrome units” and π-conjugated organic spacer led to a donor material **40** (Figure 9) of which the performance depends on the amount of acceptor selected (greater the amount of PCBM, better the performance) [78].

Under optimized conditions, an active layer composed of donor **40** exhibits relatively low performance (PCE = 0.22%), which was attributed to various factors (poor phase separation, low energy level of triplet state in the polymer, and weakly exothermic charge transfer from the singlet to PCBM). Later, the same group carried out comparative photophysical, PV and organic field-effect transistor (OFET) studies using two organometallic polymers **41** and **42** (Figure 9) [79]. These two polymers exhibited excellent absorption profiles and respectable PV performance. For instance, thiophene-containing polymer **41** showed *J*_sc_ = 3.79 mA/cm^2^, V_oc_ = 0.65 V, FF = 63.86, PCE = 1.66% while phenylene containing polymer **42** exhibited *J*_sc_ = 1.61 mA/cm^2^, V_oc_ = 0.73 V, FF = 41.37, PCE = 0.52%. Notably, polymer **41** showed a better performance and a high FF (~65%) with higher loading of the acceptor PC_71_BM. The low PCE for 35 (~0.52%) can be attributed to its blue-shifted absorption compared to **41**, as well as poor nanoscale phase separation observed with PC_71_BM in the active layer.

In a recent work, Huang and coworkers [80] compared the properties and performances of organic and organometallic random terpolymers **43** (Figure 9). Despite the fact that these two classes of materials exhibit almost similar optical properties ($E_{g}^{op}$ = 1.56 eV for organic vs. 1.53–1.56 eV for organometallic polymers), there was a marked difference in their OPV performance. For instance, the device based on donor polymer grafted with 1.5 mol% cyclometalated Pt(II) complex showed a maximum PCE of 8.45% (*J*_sc_ = 16.21 mA/cm^2^, V_oc_ = 0.80 V V, FF = 65.01%) while its organic counterpart exhibited a PCE of 7.92% (*J*_sc_ = 14.98 mA/cm^2^, V_oc_ = 0.81 V V, FF = 65.10%). This difference in performance was attributed to the higher hole mobility, limited bimolecular recombination and charge separation in the organometallic polymer. However, increasing the Pt(II) content (5 mol%) in the polymer did not enhance the performance (*J*_sc_ = 13.50 mA/cm^2^, V_oc_ = 0.81 V, FF = 66.21, PCE = 7.24%) due to its reduced charge transport.

In addition to the above-discussed donor layers in bulk heterojunction (BHJs), cyclometalated complexes have also been utilized for the fabrication of all of the polymer solar cells (all-PSCs) in which both the p-type (donor) and the n-type (acceptor) active layers are composed of the conjugated polymer. Compared to the other architectures, all PSCs are considered more suitable for large-scale applications due to their long-term optical, mechanical, thermal and morphological stabilities and fullerene-free nature. However, the development of high-performance all-PSCs is challenging due to the limited number of n-type polymers and relatively difficult control of polymer aggregation behavior [81]. It was found that the chemical conjugation of a certain amount of cyclometalated Pt(II) complex for random terpolymer acceptors facilitates exciton dissociation, while the introduction of the corresponding pure, organic main ligand exhibited the opposite phenomenon [81]. Cyclometalated Pt(II) complex containing polymers have been utilized to form an active layer (both as donor and acceptor) in all PSCs. For instance, a very first example of all-PSCs constructed using random terpolymer acceptors **44** (Figure 10) was reported by Tao and coworkers [82]. They judiciously introduced various feed ratio (1, 2, and 5 mol%) Pt complex as the third monomer into the conjugated backbone of a well-known acceptor polymer poly[[N,N′-bis(2-octyl dodecyl)-naphthalene-1,4,5,8-bis(dicarboximide)-2,6-diyl]-*alt*-5,5′-(2,2′-bithiophene)] (PNDIT2) to realize organometallic acceptor polymer.

Compared to their organic counterparts, organometallic polymers exhibited better features (viz. melting point, crystallization temperature, FMO energy levels) and PV performance (*J*_sc_ = 11.73 mA/cm^2^, V_oc_ = 0.79 V, FF = 0.49, PCE = 4.51% vs. *J*_sc_ = 10.44 mA/cm^2^, V_oc_ = 0.79 V, FF = 0.48, PCE = 3.88%) with the donor poly([2,6′-4,8-di(5-ethylhexylthienyl)benzo[1,2-*b*;3,3-*b*]dithiophene]{3-fluoro-2[(2-ethylhexyl)carbonyl]thieno[3,4-*b*] thiophenediyl}) (PTB7-Th). In this case, it was realized that the polymer with low Pt(II) content is best suited to fabricate both conventional and inverted BHJs as they possess higher exciton transport ability, well-interpenetrated nanoscale phase-separated network and efficient charge separation. However, the polymer with higher Pt(II) content (5%) showed lower performance (*J*_sc_ = 11.09 mA/cm^2^, V_oc_ = 0.79 V, FF = 0.47, PCE = 4.14%) due to poor charge transport and low optical absorption behavior. The same group [83] reported Pt(II) complex-based terpolymer acceptors **45** (Figure 10) using another well-known acceptor PNDIT2. The new acceptor polymers were realized by replacing the naphthalene diimide unit with varying amount of Pt(II) complex, (dbm)PtPyTPA (dbm = dibenzoylmethane, TPA = triphenylamine).

Like others, the new Pt(II) acceptors showed properties comparable to organic counterpart ($E_{g}^{op}$ = 1.45 eV for organic vs. 1.47–1.48 eV for organometallic polymers). Similar to the previous example, both the conventional and inverted configurations of all-PSCs showed better performance (Figure 11, values taken from Refs. [82,83]). For example, under optimized condition, a conventional BHJ device composed of 5% acceptor showed the maximum PCE of 3.97% (*J*_sc_ = 9.09 mA/cm^2^, V_oc_ = 0.77 V, FF = 0.37), which was almost double than the metal-free counterpart (*J*_sc_ = 11.09 mA/cm^2^, V_oc_ = 0.79 V, FF = 0.47, PCE = 2.58%). They also found that when the donor polymer PNDIT2 in a conventional device is replaced by another Pt-containing terpolymer donor, the PCE value drastically enhanced (PCE = 3.97% → 6.18%) A similar trend was reported for inverted BHJs, albeit with relatively low enhancement (34% higher). The higher performance was attributed to more efficient exciton separation, less charge recombination and higher hole and electron mobilities with enhanced *J*_sc_ and FF. Overall, these works have firmly established the importance of incorporating cyclometalating Pt(II) fragments to realize high-performance donor and acceptor materials.

To underpin the underlying mechanism of an enhanced PV performance, Tao and coworkers [81] assessed the PV performance of terpolymer acceptors **46** (Figure 10) prepared by embedding PyTPAPt(acac) (PyTPA = N,N′-Bis(4-bromophenyl)-4-(4-pyridinyl)benze-namine) as the third monomer into the PNDIT2. They noted that as the feed ratio of PyTPAPt(acac) complex increases from 1 to 5 and 10%, its PCE value also increased (2.49 to 2.85 and 3.16%, respectively). On the other hand, low PCE (1.85–2.47%) was found for metal-free terpolymers. They attributed these opposite PCE trends to the differences in film morphology (instead of triplet excitons in organometallic polymers) and enhanced carrier mobility, suppressed bimolecular recombination as well as improved exciton dissociation rate and charge transferability. They found that the poor molecular packing is beneficial in polymer blends while highly crystalline structure is detrimental for obtaining high-performing terpolymer acceptors suitable for all PSCs.

### 3.2. Organic Light-Emitting Diodes (OLEDs)

As discussed in the previous section, phosphorescent cyclometalated Pt(II) complex bearing materials are promising candidates for the development of OPVs. These materials have also shown potential for organic light-emitting diodes (OLEDs) due to their high PLQY, short nonradiative lifetime, and high thermal stability [5]. In addition, the square planar geometry assists in making a strong intermolecular interaction (viz. π and π* orbitals of conjugated organic ligand and Pt…Pt) between the discrete units, leading to bathochromic excimer emission. The co-existence of both the monomer and the excimer can produce a broad emission band that covers the entire visible region, resulting in white light emission that makes it possible to apply in organic lighting. High color purity and defined emission wavelengths are the added benefits offered by such materials. Using such materials, blue and white emitters can be realized, which are otherwise difficult. Complexes **21** (R_1_ = R_2_ = Me; R_3_ and R_5_ = H; R_4_ and R_6_ = F, Figure 5) and **47**–**50** (Figure 12) are some of the representative examples of emitters based on cyclometalated Pt(II) β-diketonate complexes reported for the fabrication of blue, red and white OLEDs with external quantum efficiency (EQE) 8.5–16% [84]. In addition, multimetallic complexes **28**–**37** (Figure 8) are also some of the recent examples with outstanding OLED performances. For example, OLED device based on bimetallic complex **28** (Figure 8) showed better external quantum efficiency (EQE = 0.31% at 16 wt%) and other parameters as compared to its monometallic counterpart (EQE = 0.14% at 16 wt%). OLED fabricated using complex **30** (Figure 8) exhibited a NIR emission peak at 700 nm with EQE of 6.06%. Similarly, binuclear **35** (Figure 8) and trinuclear **36** (Figure 8) emitters [64] outperformed their monometallic counterparts. Whereas complex **35** showed EQE = 10.5%, current efficiency (CE) = 21.4 cd/A, and power efficiency (PE) = 12.9 lm/W, complex **36** showed EQE = 17.0%, CE = 35.4 cd/A, and PE = 27.2 lm/W. Recently, a trinuclear complex has been reported to have EQE = 16.92%, CE = 56.74 cd/A, and PE = 29.09 lm/W [85]. Indeed, these values are among the highest reported for the multinuclear Pt(II) complex-based OLEDs.

In order to delineate the structural and electronic influences on the emission properties, Strassner and coworkers [18] reported thiazol-2-ylidene Pt(II) complexes based on the N-phenyl-4,5-dimethyl-1,3-thiazol-2-ylidene NHC ligand and seven different β-diketonate ligands. They found that complex **17** (R_1_ = R_2_ = Me; R_4_ = R_5_ = R_6_ = H; R_3_ = mes Figure 4) exhibits promising thermal stability, high quantum yields with a decay lifetime of 10.0 µs and stable emission properties in neat films. Interestingly, the emission spectra of the complex were independent of the type β-diketonate ligand selected, which is quite sensitive in other (C^C)Pt(O^O) derivatives such as imidazole or triazole NHCs containing complexes. Under optimized condition, a mixed-matrix OLED device doped with 15% of complex **17** (R_1_ = R_2_ = Me; R_4_ = R_5_ = R_6_ = H; R_3_ = mes, Figure 4) showed EQE = 12.3%, LE = 24.0 lm/W and CE = 37.8 cd/A at 300 cd/m^2^ with a green emission color. This performance was better than the previously reported C^C* cyclometalated Pt(II) complexes [29,86]. For instance, C^C* cyclometalated Pt(II) complexes **2** (R = duryl, Figure 2) based on imidazole NHC ligands and acac derivatives exhibited EQE 12.6%, LE = 11.9 lm/W and CE = 25.2 cd/A with a blue emission [29]. In such complexes, auxiliary ligands with bulky groups protect the central metal and prevent the square-planar complexes from aggregation. Methyl substituted diaryl diketonate ligands were found to be the best choice as co-ligands in combination with 1,3-diphenylbenzo[d]imidazol-2-ylidene (dpbic) as NHC ligand because they facilitate good solubility, steric protection and desirable photophysics. It was found that the presence of an additional iridium hole-transporter in the emissive layer improved the device lifetime by a factor of seven.

Recently, Kang and coworkers [54] carried out an extensive study to delineate the relationship between intermolecular interactions in the solid state and OLED performance (**23**, Figure 5). They successfully demonstrated that the incorporation of a bulky TMS group at two different positions on the 2′,6′-difluoro-2,3′-bipyridine chelate ligand is an effective strategy to enhance the PLQY. The WOLED doped with **23** (R = 5-TMS) exhibits the highest EQE = 12.3%, while the WOLEDs doped with **23** (R = none) and **23** (R = 4-TMS) exhibit similar EQEs = 7.2% and 7.6%, respectively. The low EQE of the device doped with **23** (R = none) is due to its low PLQY. The current efficiency and power efficiency are also high in the WOLEDs doped with the **23** (R = 5-TMS). In the EL spectra, all devices emit in both the blue and red wavelength regions; blue emission is caused by the intrinsic emission of the blue phosphor monomer, and the red emission is due to excimer emission. A bathochromic shift of the peak wavelength both in the blue and red emission regions were observed in the devices containing **23** (R = 4-TMS) and **23** (R = 5-TMS).

WOLEDs by utilizing excimer-based electroluminescence (EL) were also reported by using Pt(II) complexes **51** (Figure 12) bearing peripheral carbazole moieties [87]. The complexes exhibited monomer and excimer emissions in the film state, and their ratios were varied by the steric hindrance of the cyclometalated and ancillary ligands. The devices using the acac complexes exhibited predominantly excimeric EL, whereas the dipivaloylmethanate (R = ^t^Bu) complexes exhibited white EL accompanied by a high average color rendering index (CRI) of 81 due to the balanced blue monomer and orange excimer emissions Using complex **22** (R_1_ = R_2_ = F) as a single emitting dopant, a pseudo-white OLED with CIE chromaticity coordinates of (0.42, 0.42) was also fabricated that showed maximum current efficiency of 16.0 cd/A [53].

Wei et al. [88] proposed that the molecule should not only possess planar structure but also have appropriate steric hindrance, in which the planar moiety can possess good intermolecular π-π interaction and the steric hindrance affects the molecular packing, probably in favor of the external stimuli, such as pressure and grinding. On the other hand, the steric structure has a positive effect on suppressing the emission quenching in the solid state and increasing the solubility. The resulting hybrid molecules bearing square planar and bent shape units together exhibit stimuli-responsive emissions, enhanced solubility and application as OLED dopant (Figure 13). Bearing this in mind, they reported stimuli-responsive complexes **52** (Figure 12) having square planar and bent shape units separated by an oxygen atom. Results of solution-processable OLEDs indicated that **52** (O-separated) exhibits better performances with the high EQE = 17.79% and CE = 58.31 cd/A than the one without oxygen (EQE = 13.47%, CE = 38.45 cd/A). This high EQE value (~18%) is among the highest efficiency for the devices based on external-stimuli-responsive materials.

Very recently, the first examples of phosphorescent Pt(II) complexes bearing 2- and 3-(2-pyridyl)benzo[*b*]selenophenes **53** and **54** (Figure 12) have been reported [89]. Though they possess very small differences in the molecular design, these two complexes efficiently emitted green and red phosphorescence with absolute Φ = 0.52 (for green) and 0.11 (for red). When used as phosphorescent dopants for hybrid solution-processable OLED, relatively low turn-on voltages (4.2 V) and maximum EQE of 0.8% (for red device) was noted. Further optimizations of dopant concentrations and/or usage of more appropriate hosts and/or additional layers are required for getting both white EL and high OLED efficiencies based on the designed complexes.

### 3.3. Oxygen Sensing

Molecular oxygen (O_2_) is the most vital gas present in the atmosphere and being used by kingdom animalia and plantae for their sustenance and growth. Due to its par importance, O_2_ monitoring using optical sensors is an emerging area of research with wide applicability in the area of environmental monitoring, oceanography, and biology, as well as industries [90]. Usually, a luminescent oxygen-sensitive probe (OSP)/oxygen sensitizer displays a change in emission properties when it comes in contact with the gas. A number of OSPs based on transition metal complexes have been reported in the past [91]. The fact that the triplet ground state molecular oxygen (^3^O_2_) is an excellent quencher to the triplet excited-state of the metal complexes, it readily quenches the phosphorescence of the complex. Usually, cyclometalated Pt(II)/Ir(III) complexes with C^N ligands show high Φ, good photostability and long τ (in μs). The sensitivity of transition metal complexes to external stimuli and gases was exploited to create functional materials and sensors.

The development of OSPs for continuous online monitoring of molecular oxygen based on cyclometalated Pt(II) complex is well-documented [91,92]. For instance, Liu and coworkers [93] reported the very first example of a nitrile compound **55** (R = 5-CN, Figure 14) based on cyclometalating ligand for efficient O_2_ sensing. In the past, the same group [92] carried out a systematic study to understand the role of substituents on C- and N-donor fragments. They prepared a series of 2-phenylquinoline-based cyclometalated Pt(II) complexes (**56**, Figure 14) and demonstrated that the introduction of a diphenylamino group at the 4-position of the phenyl ring affects the HOMO level of the complex significantly, resulting in a marked decrease in Eg. On the other hand, the emission or energy gap is not influenced clearly by other substituents (methyl, cyano, fluoro, trifluoromethyl, methoxyl, carbazol-9-yl) at the 4-position of the phenyl ring. All the complexes show red-shifted room-temperature phosphorescence emission (at 578–599 nm) relative to the model complex Pt(ppy)acac (at 486 nm). Film containing a complex with a triphenylamino moiety, i.e., **56** exhibits the highest sensitivity ($K_{\mathrm{sv}}^{\mathrm{app}}$ = 0.020 Torr^−1^). In complex **21** (R_1_ = R_2_ = Me; R_3_ = R_4_ = H; R_5_, R_6_ = 4-C_6_H_4_F, 3,4-C_6_H_3_F_2_, 3,4,5-C_6_H_2_F_3_, Figure 5), it was noted that the presence of a bulky fluorophenyl substituents at the *para* position is beneficial to oxygen sensing [51]. Higher oxygen sensitivity was attributed to the reduced intermolecular interaction and self-quenching. In general, the following trend was observed: complexes having one fluoro-substituent was more sensitive than those having two and three fluorines. Similarly, fluorophenyl group at *para* position on the phenyl ring of 2-ppy demonstrated higher oxygen sensitivities than the *meta*-analogue. Pt(II) complex substituted at the *para* position on the phenyl ring of ppy ligand by 3,4,5-fluorophenyl showed higher luminescent oxygen sensitivities compared to its *meta*-substituted counterpart [51,52].

Very recently, Yu and coworkers [19] studied the effects of phenyl/thienyl substituents at acac auxiliary ligands on the properties of cyclometalated Pt(II) complexes **57** (Figure 14). Films prepared by immobilizing complexes in the ethyl cellulose (EC) was used as OSPs and revealed that **57** (R_1_ = CF_3_, R_2_ = NPh_2_ and R_3_ = Ph) demonstrates the highest sensitivity (Figure 15A). Moreover, the phosphorescence quenching and recovery processes of the oxygen sensing films are reversible (4000 s) and provide excellent operational stability. These results show that the phosphorescence lifetimes and oxygen sensitivity of the cyclometalated Pt(II) complexes could be tuned by the structures of β-diketone ancillary ligands.

In another study, the same group [52] assessed the effects of fluorine and phenyl substituents on oxygen sensitivity and photostability of cyclometalated Pt(II) complexes (**21**, R_1_ = R_2_ = Me; R_3_ = R_5_ = H; R_4_ = H, F; R_6_ = F, C_6_H_5_, 2,4-C_6_H_3_F_2_, Figure 5). The phosphorescence of all Pt(II) complexes in tetrahydrofuran (THF) or immobilized in EC films were gradually quenched with increased oxygen concentration (Figure 15B). It was noted that the introduction of a phenyl group or a 2,4-difluorophenyl substituent at the *para*-position on the benzene ring of 2-ppy can increase oxygen sensitivity. Complex **21** (R_1_ = R_2_ = Me; R_3_ = R_4_ = R_5_ = H; R_6_ = C_6_H_5_, Figure 5) demonstrated the highest oxygen sensitivity both in the THF solution and EC film.

On the other hand, **21** (R_1_ = R_2_ = Me; R_3_ = R_4_ = R_5_ = H; R_6_ = 2,4-C_6_H_3_F_2_, Figure 5) demonstrates the highest photostability compared to phenyl-substituted or fluorinated Pt(II) complexes. Very fast response time (3.8 s) and recovery time (4.5 s) were obtained from the **21** (R_1_ = R_2_ = Me; R_3_ = R_4_ = R_5_ = H; R_6_ = 2,4-C_6_H_3_F_2_, Figure 5)-immobilized EC film with an excellent operational stability. These results indicate that the 2,4-difluorophenyl-substituted cyclometalated Pt(II) complex is a potential OSP for photostable and fast-responsive oxygen sensing devices. In a similar and early study, Liu and coworkers [48] assessed the role of installing trifluoromethyl group at 5-position of pyridine ring on the oxygen sensing and other properties. They noted that the introduction of a strong electron acceptor group improves the oxidation potentials and PLQY of Pt(II) complexes effectively and subsequently reduces photobleaching. In addition, trifluoromethyl substituted Pt(II) complexes **58** (Figure 14) immobilized in IMPES-C are sensitive to molecular oxygen luminescence intensity changed gradually and reversibly to the O_2_ concentrations. An extremely fast response time and recovery time of oxygen sensing films were obtained in 4.0 s and 6.0 s, respectively. A very first example of a switchable AIE-active luminogen **59** (Figure 14) has been recently reported that showed excellent sensitivity towards O_2_ [94]. A complex-embedded EC polymer **59** exhibited fast quenching/recovering cycles (Figure 15C).

### 3.4. Miscellaneous

In addition to the above-discussed emerging applications of cyclometalated Pt(II) complexes, some other intriguing applications have also been demonstrated. This includes the development of biosimulated and visible light-driven photocatalysts, which have tremendous applications in the generation of clean and renewable energy (viz. hydrogen evolution reaction HER). Researchers are working towards the design of polymer dots (Pdots) in which Pt(II) complex unit is pre-synthesized as a co-monomer and then covalently linked to a conjugated polymer backbone. The resulting materials often exhibit tunable water solubility.

Chou and workers [9] demonstrated that the rate of HER could be significantly increased (up to 12×) when a cycloplatinated core is covalently attached to the Pdots (**60**, Figure 16). Increasing the mole fraction of the Pt(II)-containing unit in polymer poly[(9,9′-dioctylfluorenyl-2,7-diyl)-co-(6,7-difluoro-2,3-bis(3-(hexyloxy)phenyl)-5,8-di(thiophen-2-yl)quinoxaline)] (PFTFQ) from 0% to 15% increased the HER rate (1.3–12.7 mmolh^−1^g^−1^, Figure 17a). The performance was much better for covalently linked cycloplatinated Pdots than the physically blended counterpart Pdots (Figure 17b). It was suggested that covalent linking leads to more efficient photoinduced charge separation, as well as an enhancement in the rate of the proton reduction reaction. Furthermore, this system also holds promise as it works under a methanol-free condition, owing to its excellent water dispersibility.

To underpin the importance of acceptor comonomers on HER, they systematically studied the effect of cycloplatinated Pdots with various acceptor co-monomers [95]. It was concluded that the introduction of sp^2^ and sp^3^ N-containing acceptor co-monomer into cycloplatinated Pdots dramatically enhances the photocatalytic performance and stability. For instance, polymer dots (Pdots) based on **61** (Figure 16) Pdots provide the excellent HER (up to 7.34 ± 0.82 mmol h^−1^ g^−1^) and enhance performance (9 h) with hydrogen productions of 31.54 mmolg^−1^ compared to the pristine Pdots. The catalysis was carried out under a methanol-free solution and visible light-driven system and showed excellent reusability. It was also shown that the covalent linking strategy is better in terms of toxicity reduction as it poses less side effects than the system containing Pt nanoparticles into a solution system. Some other reports are also available on the applications of cyclometalated Pt(II) complexes in the area of data security storage [96], nonvolatile photomemories [97], NLOs [98], light electrochemical cells [63], etc. For example, it has been reported that one of the isomers of complex **62** (Figure 16) shows responsiveness (luminescence switching) against ethanol vapor in a ground state [96].

## 4. Conclusions and Outlook

In this review, we highlighted the structural features, photophysical properties, and applications of cyclometalated Pt(II) complexes bearing bidentate cyclometalating ligands and β-diketonate ancillary ligands, i.e., (C^N)Pt(O^O) and (C^C)Pt(O^O) derivatives. Benefitting from the heavy metal effect of platinum and the strong field nature of the cyclometalating ligands, this class of compounds offers easy synthetic pathways, bright and persistent luminescence, stable and strong phosphorescence, wide scope for structural modification, etc. Using pertinent examples, we showed that new functional materials could be realized by a judicious interplay between covalent and noncovalent interactions as well as altering molecular separation/stacking, which is achievable via ancillary as well as cyclometalating ligands. Overall facile structural modification, good water dispersivity, tunable optoelectronic properties make this class of compounds really impressive.

In terms of applications, we have shown that these materials are potential candidates for O-E devices. Several new and highly efficient materials have been developed in the last few years, having the potential to replace fullerene (an acceptor) from the OPVs and single dopants for OLEDs. Moreover, these complexes/units are being installed over large macromolecules (polymers) and converted to Pdots to be used as photocatalysts for the production of molecular hydrogen, which is in high demand and a cleaner source of energy. The newly reported photocatalysts have been found to be safe in nature, stable and reusable. Some researchers suggested their applicability for the development of nonvolatile memory and data security storage. Owing to their multicolor temperature-responsive emissions [49,50], they are also potential candidates for temperature sensing and nanophotonics.

Future work includes, but is not limited to, the exploration of newly synthesized ligands (viz. synthesis of bulky diketones and extended cyclometalating ligands), synthesis of homo- and hetero multi-metallic complexes, etc. Due to their extended absorption and emission features coupled with low toxicity, they are potential candidates for bioimaging. Especially complexes sensitive to oxygen are particularly interesting for hypoxic imaging tumors. Various studies have demonstrated that the PV performance can be drastically improved, if not all times, by changing the donor/acceptor active layers. Therefore, those materials that have been used as acceptors and shown moderate efficiency in all PSCs should be analyzed with different donor polymers.

## Figures and Tables

**Figure 1 materials-14-04236-f001:**
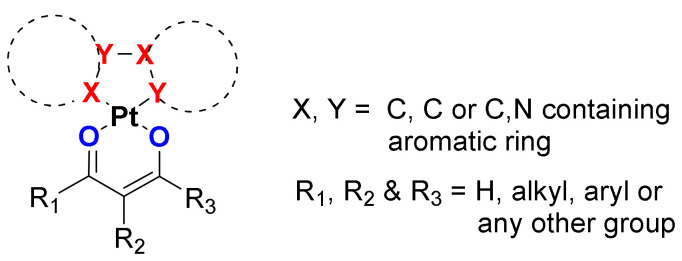
A general chemical structure of cyclometalated Pt(II) β-diketonate complexes covered in this review. Two aromatic rings (dotted lines) are joined through C-C or C-N bond.

**Figure 2 materials-14-04236-f002:**
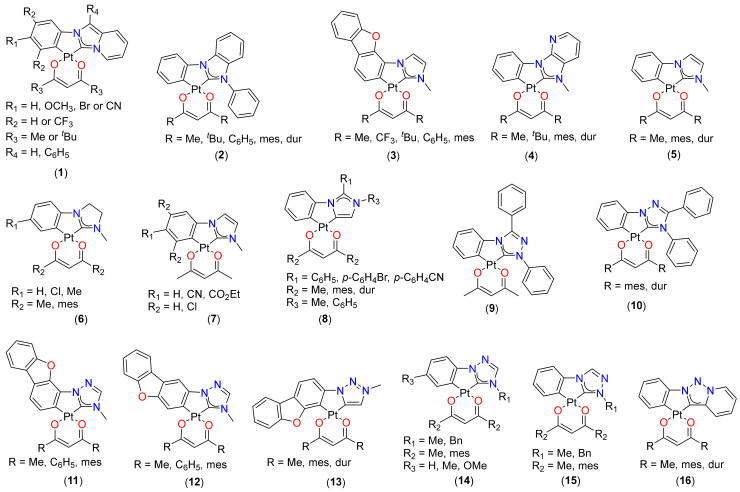
Pt(II) complexes based on imidazole, pyrazole, and triazole-derivatives based NHCs as the cyclometalating ligand and β-diketonate derivatives as ancillary ligands.

**Figure 3 materials-14-04236-f003:**
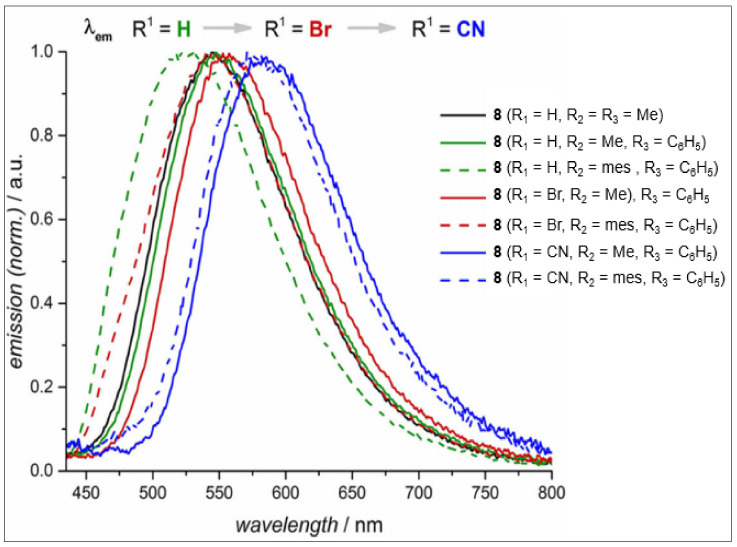
Emission spectra of complex **8** in PMMA matrix (2 wt% emitter load, λ_exc_ = 360 nm) at room temperature. Reproduced with permission from ref. [34].

**Figure 4 materials-14-04236-f004:**
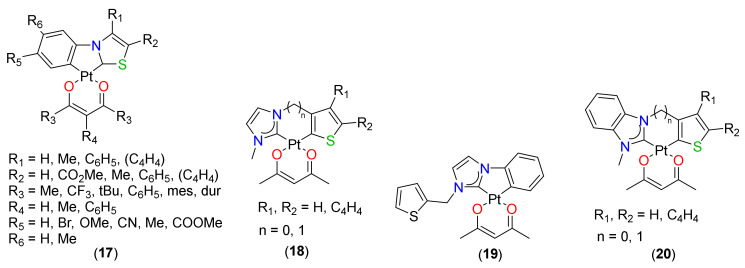
Pt(II) complexes based on thiazole-derived NHCs as the cyclometalating ligand and β-diketonate derivatives as ancillary ligands.

**Figure 5 materials-14-04236-f005:**
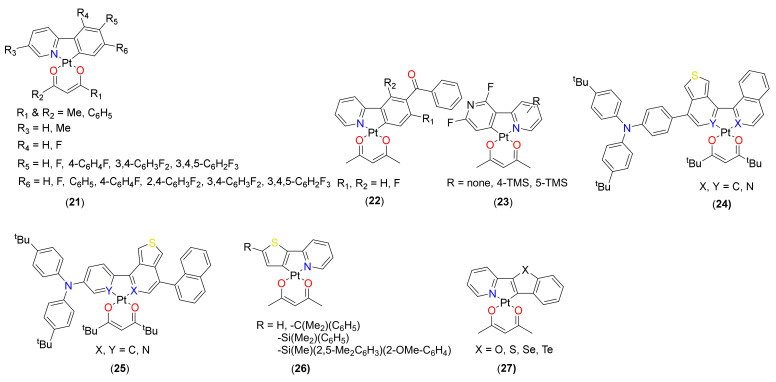
Pt(II) complexes based on 2-phenylpyridine, 2-thienylpyridine, 2-phenylbenzimidazole, etc. cyclometalating ligands and β-diketonate derivatives as ancillary ligands.

**Figure 6 materials-14-04236-f006:**
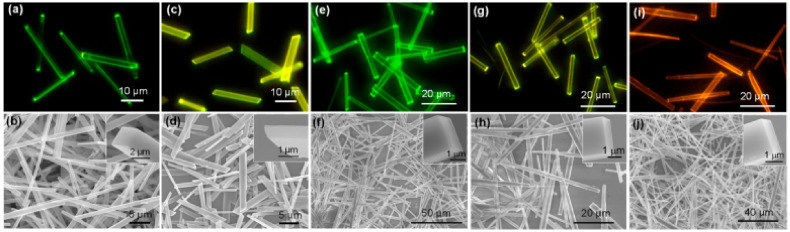
(**a**,**c**,**e**,**g**,**i**) Fluorescence microscopy images and (**b**,**d**,**f**,**h**,**j**) SEM images of microcrystals of (**a**,**b**) **21-g** (R_1_ = R_2_ = C_6_H_5_; R_3_ = Me; R_4_ = R_6_ = F; R_5_ = H) (**c**,**d**) **21-y** (R_1_ = R_2_ = C_6_H_5_; R_3_ = Me; R_4_ = R_6_ = F; R_5_ = H) (**e**,**f**) **21** (R_1_ = R_2_ = C_6_H_5_; R_3_ = Me; R_4_ = F; R_5_ = R_6_ = H), (**g**,**h**) **21** (R_1_ = R_2_ = C_6_H_5_; R_3_ = Me; R_4_ = R_6_ = H; R_5_ = F, and (**i**,**j**) **21** (R_1_ = R_2_ = C_6_H_5_; R_3_ = Me; R_4_ = R_5_ = H; R_6_ = F). The inset shows enlarged SEM images of the ends of microrods. Reproduced with permission from ref. [49].

**Figure 7 materials-14-04236-f007:**
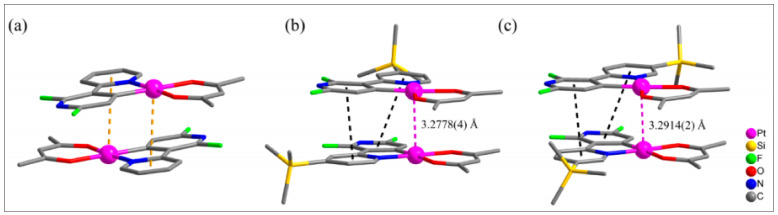
Dimeric structures of complexes **23** (R = none, 4/5-TMS) formed via Pt···π (yellow-dashed lines), π−π (black-dashed lines), and Pt···Pt (violet-dashed lines) interactions. H atoms were omitted for clarity. (**a**) R = none; (**b**) R = 4-TMS and (**c**) R = 5-TMS. Reproduced with permission from ref. [54].

**Figure 8 materials-14-04236-f008:**
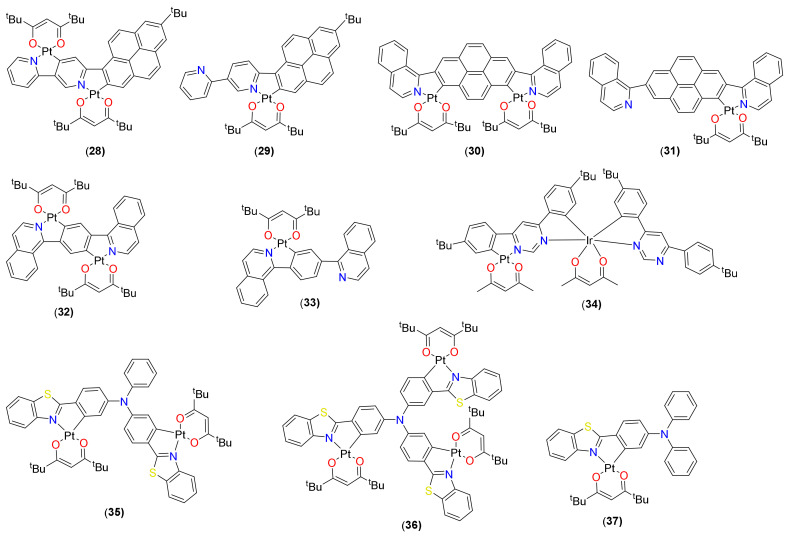
Mono-, di- and multinuclear Pt(II) complexes with excellent PL and OLED performances.

**Figure 9 materials-14-04236-f009:**
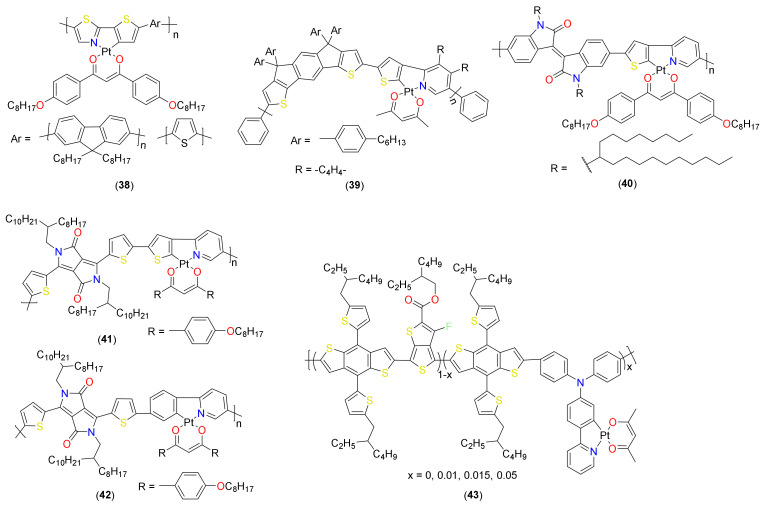
Cyclometalated Pt(II) complexes-based donor materials.

**Figure 10 materials-14-04236-f010:**
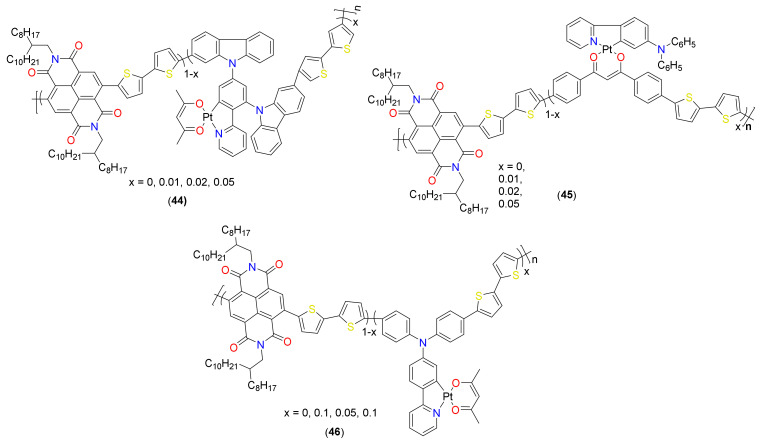
Cyclometalated Pt(II) complexes-based acceptor polymers for all-PSCs.

**Figure 11 materials-14-04236-f011:**
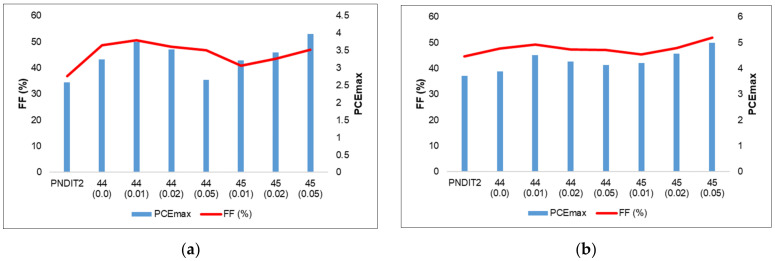
Variation of FF and PCE in (**a**) conventional (ITO/PEDOT:PSS/PTB7-Th:**44**–**45**/PDIN/Al) and (**b**) inverted (PTB7-Th:**44**–**45**) cells with Pt(II) content. Value given in the parenthesis indicates certain amount (%) of Pt(II) cyclometalated complex.

**Figure 12 materials-14-04236-f012:**
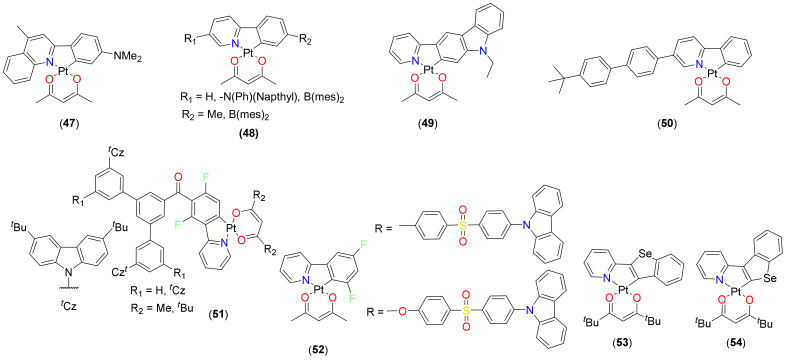
Cyclometalated Pt(II) complexes-based luminophores for OLEDs.

**Figure 13 materials-14-04236-f013:**
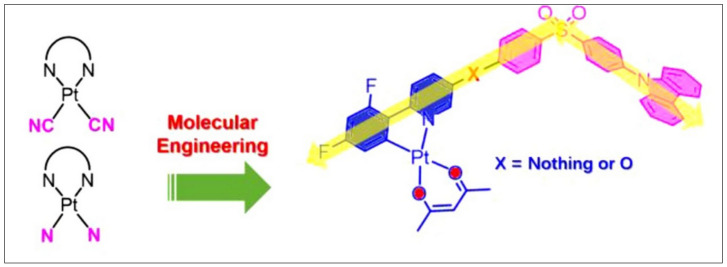
Molecular engineering scheme for the realization of external-stimuli-responsive OLED phosphors. Reproduced with permission from ref. [88].

**Figure 14 materials-14-04236-f014:**
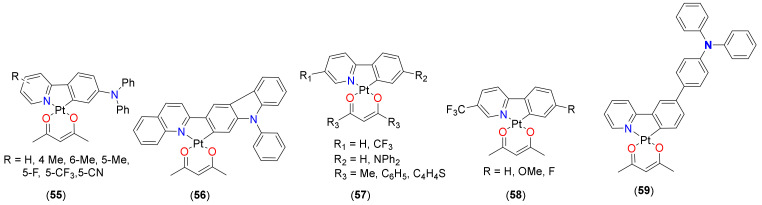
Cyclometalated Pt(II) complex-based oxygen sensors.

**Figure 15 materials-14-04236-f015:**
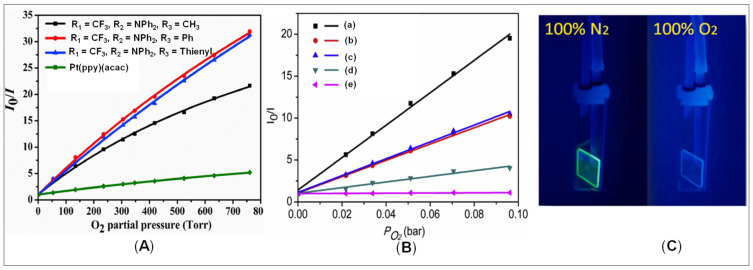
Stern Volmer plots of (**A**) complex **57** immobilized on EC films, (**B**) complex **21** in THF (a: R_1_ = R_2_ = Me; R_3_ = R_4_ = R_5_ = H; R_6_ = C_6_H_5_, b: R_1_ = R_2_ = Me; R_3_ = R_4_ = R_5_ = H; R_6_ = 2,4-C_6_H_3_F_2_, c: R_1_ = R_2_ = Me; R_3_ = R_4_ = R_5_ = R_6_ = H, d: R_1_ = R_2_ = Me; R_3_ = R_4_ = R_5_ = H; R_6_ = F, e: R_1_ = R_2_ = Me; R_3_ = R_5_ = H; R_4_ = R_6_ = F) and (**C**) The luminescence photographs of **59** film under 100% N_2_ and 100% O_2_ (excited at 365 nm). Reproduced with permission from ref. [19,52,94].

**Figure 16 materials-14-04236-f016:**
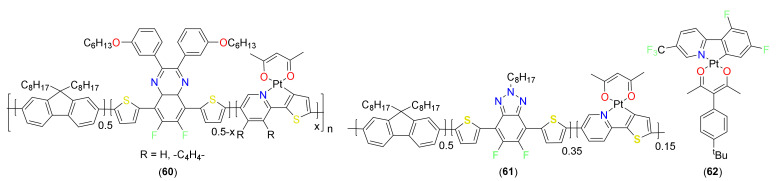
Cyclometalated Pt(II) complex-based photocatalysts and others.

**Figure 17 materials-14-04236-f017:**
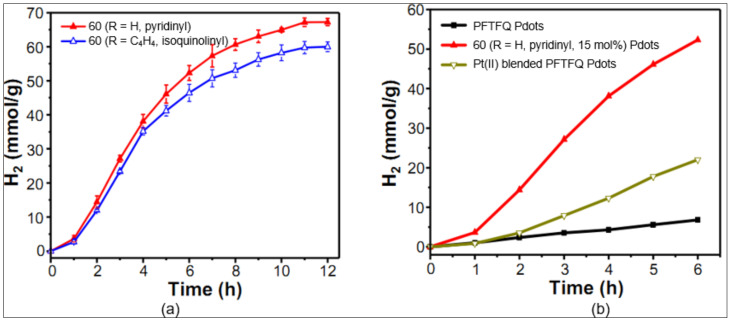
(**a**) Time course of produced H_2_ for **60** (15 mol%) Pdots for 12 h. (**b**) Comparison of hydrogen generation of chemically linked **60** Pdots and physically Pt(II)-blended-counterpart PFTFQ Pdots. Reproduced with permission from ref. [9].

## Data Availability

No new data were created in this study.

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
