# Peer review of "Functional Materials Based on Cyclometalated Platinum(II) β-Diketonate Complexes: A Review of Structure–Property Relationships and Applications"

_materials, 2021, doi:10.3390/ma14154236_

Round 1

Reviewer 1 Report

Authors proposed a review paper entitled: “Functional materials based on cyclometalated platinum(II) β-diketonate complexes: A review of structure property relation-ships and applications”. This paper has been proposed for the Processes Journal, MDPI.

A good use of English has been employed in this manuscript.

This paper has a good scientific soundness.

Authors have studied the literature deeply.

I recommend minor revisions, as in this list has been stated:

Line 33. dot after the references.

Line 47. Ф and τ values not defined. ФT and τT defined previously.

Line 60. a reference could be requested here.

Line 119. did you define mes?

Line 145. “2% in PMMA”. is this percentage on mass basis?

Line 154. “acetylacetonate (acac )”. maybe better defined with capital letters, this acronym.

I suggest adding an abbreviation list, according to the guidelines of this Journal.

Also, not all the acronyms used in this paper have been defined the first time that they were used. I suggest to check this in the manuscript.

Line 412. is this widely supported by literature? “The incorporation of Pt(II) fragment 412 significantly red-shifted the absorption, lowered the frontier orbital energy levels and im-413 proved the solar cell performance compared to metal-free polymers”

Line 422. “PV and OFET studies” are these defined?

Figure 5. do you think that it could be useful and scientifically valid, adding error bars to this diagram?

Figure 6. the focus of this figure needs to be ameliorated, especially regarding the written information in the figure. maybe the authors could consider to move part of this written information to the caption of the figure.

Figure 7a. try to ameliorate the landscape of this figure, while reproducing it.

Line 709. “comonomers on HER ,”. The space among HER and the comma should be eliminated.

Author Response

Response to Reviewer # 1

Authors proposed a review paper entitled: “Functional materials based on cyclometalated platinum(II) β-diketonate complexes: A review of structure property relation-ships and applications”. This paper has been proposed for the Processes Journal, MDPI. A good use of English has been employed in this manuscript. This paper has a good scientific soundness. Authors have studied the literature deeply. I recommend minor revisions, as in this list has been stated:

 Comment 1:      Line 33. dot after the references.

Response:           We thank the reviewer for appreciating our review and recommending it for the publication. As suggested, dot has been placed after the references.

Comment 2:       Line 47. Ф and τ values not defined. ФT and τT defined previously.

Response:           Ф and τ values have been defined in the revised MS.

Comment 3:       Line 60. a reference could be requested here.

Response:           A suitable reference has been cited to support the sentence.

Comment 4:       Line 119. did you define mes?.

Response:           Yes, mes was already defined (line 108).

Comment 5:       Line 145. “2% in PMMA”. is this percentage on mass basis?.

Response:           2% in PMMA is 2 wt% in PMMA. We have updated this throughout the MS.

Comment 6:       Line 154. “acetylacetonate (acac )”. maybe better defined with capital letters, this acronym.?

Response:           Throughout the MS, we abbreviated acetylacetonate as “acac”. This acronym is widely used in the literature.

Comment 7:       I suggest adding an abbreviation list, according to the guidelines of this Journal.

Response:           Abbreviations are defined the first time they appear following the guidelines of this Journal.

Comment 8:       Also, not all the acronyms used in this paper have been defined the first time that they were used. I suggest to check this in the manuscript.

Response:           Please see response to comment 7.

Comment 9:       Line 412. is this widely supported by literature? “The incorporation of Pt(II) fragment 412 significantly red-shifted the absorption, lowered the frontier orbital energy levels and im-413 proved the solar cell performance compared to metal-free polymers.

Response:           This is often observed, but it is not always true that the incorporation of metal fragment would improve the solar cell performance. An exhaustive account on this issue can be found in references cited in the MS.

Comment 10:     Line 422. “PV and OFET studies” are these defined?.

Response:           PV and OFET have been defined in the revised MS.

Comment 11:     Figure 5. do you think that it could be useful and scientifically valid, adding error bars to this diagram?.

Response:           We plotted Figure 5 using the exact data reported in reference. Due to the lack of error data, we preferred to plot the graph using the same values.

Comment 12:     Figure 6. the focus of this figure needs to be ameliorated, especially regarding the written information in the figure. maybe the authors could consider to move part of this written information to the caption of the figure..

Response:           Fig. 6 has been modified as suggested and part of the written information has been moved to the main text.

Comment 13:     Figure 7a. try to ameliorate the landscape of this figure, while reproducing it.

Response:           Fig. 7a has been modified as suggested.

Comment 14:     Line 709. “comonomers on HER ,”. The space among HER and the comma should be eliminated..

Response:           It has been corrected in the revised MS.

Reviewer 2 Report

The manuscript entitled "Functional materials based on cyclometalated platinum(II) β-diketonate complexes: A review of structure property relationships and applications", sent for publication in Materials is appropriate for this journal. It presents a review on a subject of high interest both for scientific and medical fields.

Author Response

Response to Reviewer # 2

The manuscript entitled "Functional materials based on cyclometalated platinum(II) β-diketonate complexes: A review of structure property relationships and applications", sent for publication in Materials is appropriate for this journal. It presents a review on a subject of high interest both for scientific and medical fields.

Response:           We thank the reviewer for appreciating our article and recommending it for publication.

Reviewer 3 Report

Authors describe in a review an extensive study about Pt(II) planar complexes (with two different bidentate ligands, one being acac. Those complexes have been described from PL properties mainly.
It is a very huge collection about a thematic where the authors published some articles.

The review is interesting but should be rearranged for simplicity.

All charts should be simplified so that the number of drawn structures could be minimized (and not corresponding to (one structure drawn = one article). Very often, some structures could be merged.

Some images do not have high dpi and seem blurry. (better resolution is required.

Then, adding some tables where the relationship between structure and parameters (Φ and τ) would be welcome.

Finally, authors should check the references and add the right abbreviations for the cited articles.

Author Response

Response to Reviewer # 3

Authors describe in a review an extensive study about Pt(II) planar complexes (with two different bidentate ligands, one being acac. Those complexes have been described from PL properties mainly. It is a very huge collection about a thematic where the authors published some articles. The review is interesting but should be rearranged for simplicity.:

Comment 1:       All charts should be simplified so that the number of drawn structures could be minimized (and not corresponding to (one structure drawn = one article). Very often, some structures could be merged.

Response:           We thank the reviewer for appreciating our article. As suggested, some structures have been merged and the overall number of structures have been reduced.

Comment 2:       The format in drawing and references are inconsistent; please check. 

Response:           We have thoroughly checked the MS and revised it complying with the journal’s format. Changes have been made in Tables, Figs. and Refs. wherever required to avoid inconsistency.

Comment 3:       Then, adding some tables where the relationship between structure and parameters (Φ and τ) would be welcome.

Response:           As suggested, we have added extra Tables in depicting photo-physical parameters viz. wavelength, quantum yield, efficiency.

Comment 4:       Finally, authors should check the references and add the right abbreviations for the cited articles.

Response:           We have thoroughly checked and edited the references as per the journal requirements.

Reviewer 4 Report

The authors have systematically summarized the works on luminescent cyclometalated platinum(II) complexes with β-2 diketonate ligands from the literature. A vast number of examples have been covered, with brief explanations on their properties. The examples are grouped under a few categories according to their applications and are well-organized for the readers to follow.

At the same time these review reads far more like an annotated bibliography of a large number of results as opposed to the sort of critical assessment of a field that we look for in a manuscript of this type. The authors go to great lengths to summarize what has been observed, but a review should be more than just a summary of results.  I would have liked to see more analysis of the results of previous works and discussion of the advantages and disadvantages of complexes with β-2 diketonate ligands over other ones. In addition, the authors are recommended to address the following points prior to further consideration by Materials.

  1. Why did the authors skip some recent work, for example Inorg. Chem. 59 (2020) 9308?
  2. The logic of compounds numbering is not clear. Why do some of the examples have alphabetic characters while others do not?
  3. Complexes with NHCs, viz. 1, 4, 5a, 6, 9, 10, 11, 12, and other require a symbol of pi bond delocalization.
  4. Authors should introduce an abbreviated section. For example, the abbreviation PMMA is used for the first time on page 3, while the decoding is given only on page 5. What is "BHJs" (P.12. line 445)?
  5. P. 2, line 56. It is not clear what the authors mean by "increasing the number of metal (platinum) fragments". Forming intermolecular contacts and arranging molecules in stacks? Multinuclear molecules?
  6. P.5, line 198. The authors write that the thiazole rings were functionalized, but do not write exactly how.
  7. P.7, lines 267-269. The authors write that "Indeed, the presence of fluorophenyl substituent led to reduced metallophilic (Pt ... Pt) interaction in the solid state." It is not clear how the presence of fluorine in the cyclometallated ligand leads to a decrease in the platinum-platinum interactions. The authors point to a red shift when fluorine is located in the para position, compared to the meta position, why is this happening?
  8. P. 7, lines 282-283. How does the type and position of substituents affect of emission properties? Is there a change in orbitals, electron density, or somethings other? "These results suggests" - It is not clear.
  9. P.10 lines 363-364. What is the structure of the “mono-metallic complex”?
  10. .17, lines 633-645. Again, why fluorine should be in the para position? Why are the observed results worse for the meta than for the para position? The authors write "In general, the following trend was observed: complexes having one fluoro-substituent was more sensitive than those having two and three fluorines". What could be a reason for this fact?

Author Response

Response to Reviewer # 4

The authors have systematically summarized the works on luminescent cyclometalated platinum(II) complexes with β-2 diketonate ligands from the literature. A vast number of examples have been covered, with brief explanations on their properties. The examples are grouped under a few categories according to their applications and are well-organized for the readers to follow. At the same time these review reads far more like an annotated bibliography of a large number of results as opposed to the sort of critical assessment of a field that we look for in a manuscript of this type. The authors go to great lengths to summarize what has been observed, but a review should be more than just a summary of results.  I would have liked to see more analysis of the results of previous works and discussion of the advantages and disadvantages of complexes with β-2 diketonate ligands over other ones. In addition, the authors are recommended to address the following points prior to further consideration by Materials:

Comment 1:       Why did the authors skip some recent work, for example Inorg. Chem. 59 (2020) 9308?.

Response:           We thank the reviewer for the critical assessment and reading. We also welcome suggestion to analyse the previous works and discuss their advantages/disadvantages. We have included some additional recent work in the revised MS, which were inadvertently missed.

Comment 2:       The logic of compounds numbering is not clear. Why do some of the examples have alphabetic characters while others do not?.

Response:           We have revised the numbering scheme used in the MS. Also, some structures have been merged and the overall number of structures reduced.

Comment 3:       Complexes with NHCs, viz. 1, 4, 5a, 6, 9, 10, 11, 12, and other require a symbol of pi bond delocalization.

Response:           The reviewer is right. The mistake has been corrected in the revised MS.

Comment 4:       Authors should introduce an abbreviated section. For example, the abbreviation PMMA is used for the first time on page 3, while the decoding is given only on page 5. What is "BHJs" (P.12. line 445)?

Response:           Following the journal guidelines, abbreviations are defined the first time they appear.

Comment 5:       P. 2, line 56. It is not clear what the authors mean by "increasing the number of metal (platinum) fragments". Forming intermolecular contacts and arranging molecules in stacks? Multinuclear molecules?

Response:           We meant here the number of metal units. Various studies showed that as the number of Pt(II) unit increases in the metalla-ynes or orthometalated Pt(II) complexes, the SOC enhances.

Comment 6:       P.5, line 198. The authors write that the thiazole rings were functionalized, but do not write exactly how.

Response:           It has been clarified in the revised MS.

Comment 7:       P.7, lines 267-269. The authors write that "Indeed, the presence of fluorophenyl substituent led to reduced metallophilic (Pt ... Pt) interaction in the solid state." It is not clear how the presence of fluorine in the cyclometallated ligand leads to a decrease in the platinum-platinum interactions. The authors point to a red shift when fluorine is located in the para position, compared to the meta position, why is this happening?..

Response:           It is well documented that in such complexes, the Pt...Pt distances exceed the sum of van der Waals radii for the platinum atom. Therefore, Pt...Pt distances are generally found beyond the normal range (2.7−3.5 Å), thereby excluding the possibility of interaction between two platinum metal atoms in solid state. Since fluorophenyl substituent provides steric protection around the metal centre, it might contribute to enhance the metal-metal distance. Regarding the red-shifted emission peaks, comparison was made against the parent complex Pt(ppy)(acac). We have modified the sentence in the revised MS.

Comment 8:       P. 7, lines 282-283. How does the type and position of substituents affect of emission properties? Is there a change in orbitals, electron density, or somethings other? "These results suggests" - It is not clear.

Response:           This sentence has been revised for better clarity.

Comment 9:       P.10 lines 363-364. What is the structure of the “mono-metallic complex”?.

Response:           Chemical structures of the mono-metallic complexes have been included in the revised MS.

Comment 10:     P.17, lines 633-645. Again, why fluorine should be in the para position? Why are the observed results worse for the meta than for the para position? The authors write "In general, the following trend was observed: complexes having one fluoro-substituent was more sensitive than those having two and three fluorines". What could be a reason for this fact?

Response:           Higher oxygen sensitivity by such complexes is attributed to the fluorophenyl substituent which reduces the intermolecular interaction of the complexes and prevents self-quenching compared to Pt(ppy)(acac). However, no justification was given for the observation why para was more effective than meta and why one fluoro-substituent was more sensitive than those having two and three fluorines

Round 2

Reviewer 4 Report

In my opinion the manuscript has improved significantly and it is now good for publication.